# Climate change metrics: bridging IPCC AR6 updates and dynamic life cycle assessments

Vladimir Zieger<sup>1</sup>, Thibaut Lecompte<sup>1</sup>, Simon Guihéneuf<sup>1</sup>, Yann Guevel<sup>1</sup>, Manuel Bazzana<sup>2</sup>, Thomas Gasser<sup>3</sup>, Yue He<sup>3</sup>

<sup>1</sup>IRDL, CNRS, Université Bretagne Sud, F-56100 Lorient, France

<sup>2</sup>Centre Scientifique et Technique du Bâtiment, F-38400 Saint-Martin-d'Hères, France

Correspondence to: Vladimir Zieger (vladimir.zieger@univ-ubs.fr)

Abstract. Climate change metrics result from analytical simplification of complex and diverse climate models that are generally not deeply investigated by Life Cycle Assessment communities. We investigated the Sixth Assessment Report of the Intergovernmental Panel on Climate Change to properly gather updated metric equations, climate parameters and associated uncertainties. Metrics are mainly designed with a single gas pulse emission at  $t_0$  whereas multi-gas and multi-time pulse emissions are mostly encountered in LCA modelling. Therefore, common static and relative metrics might not suit dynamic climate change assessments (dCCA) that differentiate pulse timing and gas contributions over time. This study focuses on absolute and dynamic metrics – cumulative radiative forcing (AGWP or  $\Delta F$ ) and global temperature change (AGTP or  $\Delta T$ ) – applied to well-mixed greenhouse gases. Cumulative radiative forcing assessment at 20, 100, 500 years appears sufficient. Global temperature change metrics have some advantages that offset their higher uncertainties. (1) Degree Celsius unit better suits peak warming targets. (2) Positive and negative peaks, as well as long-term temperature change, partly alleviate the time horizon decision issue while assessing product systems. (3) Graphical representations are comparable to simultaneously depict short- and long-lived climate forcers. In future assessment reports, IPCC is invited to recall climate equations and updated parameters values in a pedagogical way and to adopt peak and long-term temperature change metrics. dCCA recommendations are to plot  $\Delta F$  and  $\Delta T$  temporal profiles of product systems up to 600 years and use suggested metrics. This should enable going towards climate neutrality with more clarity, transparency and understanding.

## 1 Introduction

10

Human activities have now clearly put the Earth system well beyond the safe operating space for humanity (Richardson et al., 2023). A systemic framework on the Earth system trends (Rockström et al., 2009; Steffen et al., 2015) is essential to capture levels of anthropogenic perturbation and develop a strategy in order to maintain stability and resilience of the Earth system as a whole. Global warming is one hidden cost of any human activity emitting greenhouse gases (GHGs). Recent changes are rapid, intensifying, and unprecedented over thousands of years (IPCC, 2021a).

<sup>&</sup>lt;sup>3</sup>International Institute for Applied Systems Analysis, A-2361 Laxenburg, Austria

The study of Earth's climate considers complex interactions between Atmosphere, Biosphere, Cryosphere, Hydrosphere and Lithosphere. The number of forcing mechanisms (e.g. GHGs, aerosols) is large (Szopa et al., 2021), as are uncertainties. State-of-the-art of Earth system models used in Model Intercomparison Projects (MIPs) attempt to describe all climate system's components as accurately as possible. From MIPS, simplified parametric models are developed. In each new assessment report, the Intergovernmental Panel on Climate Change (IPCC) updates climate parameters and metrics based on recent modelling as well as on changes in background conditions. Lastly, IPCC gathers updated characterisation factors (CF)\* (\* means see definition in Appendix A: Glossary) and associated uncertainties for several metrics.

30


Global Warming Potential (GWP) has been widely used since the Kyoto Protocol thanks to its ease of calculation and simple definition, kilogram CO<sub>2</sub>-equivalent (kgCO<sub>2</sub>e) at a 100-year time horizon (H) being now a common metric to assess carbon footprint of products and systems. GWP has also been largely criticised. 1) It does not explicitly represent the temperature response to a GHG emission (Shine et al., 2005), i.e. it is a poor indicator of peak warming (Allen et al., 2022; Fuglestvedt et al., 2018). 2) There is a nonlinear relationship between integrated radiative forcings of CO<sub>2</sub> and of the studied gas (O'Neill, 2000). 3) It is statically expressed, H being a value judgement that has a decisive influence on the metric values (Myhre et al., 2013a) from both GHG emissions and temporary biogenic carbon storage (Levasseur et al., 2012). Global Temperature change Potential (GTP) is explicitly linked to temperature change (Shine et al., 2005), but remains relative to CO<sub>2</sub> and is not dynamically used by the LCA community.

Due to the variety of emitted components' physical properties and of applications, LCA studies would benefit from moving away from single metric studies towards a multi-metric perspective and sensitivity tests, combined with a careful exchange with the end-users of LCA (Levasseur et al., 2016a). Extensive efforts have been done in that sense through the UNEP/SETAC Life Cycle Initiative (Levasseur et al., 2016b). Levasseur et al. (2016a) summarized all issues one has to have in mind before choosing a climate metric. They offered the same relevant recommendations for static LCA as Cherubini et al. (2016), i.e. the use of GWP20 for short-term and GTP100 for long-term perspectives in order to complete the usual GWP100 mid-term results. Yet, these CO<sub>2</sub>-equivalent metrics have a poor temporal correspondence with temperature responses for short-lived climate forcers\* (SLCFs), i.e. are a poor indicator of temperature stabilisation (Allen et al., 2018b). Other approaches, such as the dynamic Life Cycle Analysis (dLCA) developed by Levasseur et al. (2010), consist in accounting for the timing of GHG storage and emission on a year-by-year basis, and assess them using dynamic climate metrics\*. As other impact categories than climate change are not considered here, dynamic climate change assessment (dCCA) will be used instead of dLCA throughout the rest of this paper. Hence, dCCA is based on a multi-pulse and multi-GHGs framework, as well as on absolute and dynamic climate impact assessment methods.

Limitations for a broader use of dCCA have been identified. First, IPCC does not provide the needed information in its latest report to easily understand and use dynamic climate metrics. Second, such metrics have been ignored in recent LCA final recommendations (FAO, 2023; Jolliet et al., 2018; Levasseur et al., 2016b). More deeply, the way most climate metrics and CFs are designed, i.e. based on single gas emission at time zero (t<sub>0</sub>) or on aggregated emissions and removals into one CO2-equivalent pulse at t<sub>0</sub>, may not be suitable for dCCA applications. For instance, in assessments of long-lasting systems

or materials containing biogenic carbon, multi-gas pulse emissions might happen several decades after t<sub>0</sub>. Pros and cons of addressed dynamic metrics are discussed here to support the interpretation of dynamic assessments with several emission pulses spread over time.

For that interdisciplinary research purpose, we found it useful to merge dynamic climate metrics, climate parameters and associated uncertainties, using updates from IPCC 6<sup>th</sup> Assessment Report (AR6). Special emphasis is given to two metrics: Absolute Global Warming Potential (AGWP) (or cumulative Radiative Forcing (ΔF)), an integrated metric, and Absolute Global Temperature change Potential (AGTP) (or Global Mean Temperature Change (ΔT)), an endpoint metric. According to AR6, carbon dioxide (CO<sub>2</sub>), methane (CH<sub>4</sub>) and nitrous oxide (N<sub>2</sub>O) are the three most important GHGs, responsible for 82% of positive effective radiative forcing (ERF) since the beginning of the industrial revolution (Szopa et al., 2021). The present study thus focuses on them. Given the openness of the IPCC to revise emission metrics in future Assessment Reports (Abernethy and Jackson, 2022), the proposed framework could help in selecting more robust time-dependent emission metrics and new CF for product system assessments. To sum up, this article aims:

- to give an overview of what underpins climate metrics and associated uncertainties using AR6;
- to offer AGWP and AGTP for CO<sub>2</sub>, CH<sub>4</sub> and N<sub>2</sub>O to environmental assessment communities;
- to discuss to what extent absolute and dynamic metrics  $\Delta F$ ,  $\Delta T$  better reflect product system mitigation objectives;
- to suggest clearer data presentation and new CFs in future IPCC reports that better suit single- and multi-time emission profiles for both short- and long-lived well-mixed GHGs;
- to make available an open-source dynamic climate change assessment tool that includes climate-carbon feedbacks.

## 2 Methodology


#### 2.1 Climate change metrics – background information

Emission metrics aim to provide an 'exchange rate' in multi-component policies or in areas such as LCA (Aamaas et al., 2013; Myhre et al., 2013a). Among other attempts (Edwards and Trancik, 2022; Kandlikar, 1996), Eq. (1) gives a general formulation of an absolute emission metric (AM) (Forster et al., 2007). This proposal is suited to a wide range of metrics, but not all, e.g. cost-effective metrics (Tanaka et al., 2021).

$$AM_i = \int_0^\infty \{ (I(\Delta C_{r+i}(t)) - I(\Delta C_r(t)))g(t) \} dt, \qquad (1)$$

where  $I(\Delta C_r(t))$  is the function describing the "impact" of a change in climate  $\Delta C$  (e.g., concentration, temperature) at time t, with a discount function, g(t), and compared to a reference system, r, on which the perturbation i occurs. In emission metrics, g(t) is mostly a step-function to represent a fixed time-horizon in integrated metrics, or a Dirac delta function that removes the integral of Eq. (1) to represent an instantaneous evaluation in endpoint metrics (Peters et al., 2011a). We can notice that the latter have nothing to do with LCA endpoint modelling (Bare et al., 2000): an endpoint climate metric is a

midpoint LCA indicator. To get a common scale, metrics can be given in relative terms by normalising to a reference gas:  $M_i = AM_i/AM_j$ . For instance, GWP and GTP are defined by normalising respectively AGWP or AGTP from a pulse emission of a specific GHG to respectively AGWP or AGTP of 1kg of CO<sub>2</sub>. All metrics require input parameters (Hodnebrog et al., 2013) influenced over time by changing background information (see supplementary material S1).

Integrated or sustained temperature change metrics (iAGTP, iGTP, SAGTP, SGTP) that reflect for instance sea-level rise (Sterner et al., 2014) are not considered here since they have similar behaviours as integrated radiative forcing metrics (AGWP, ΔF, GWP), at least for long-lived climate forcers (Azar and Johansson, 2012; Collins et al., 2020; Levasseur et al., 2016a). Recently developed step/pulse metrics have been proposed to better include SLCFs (Dhakal et al., 2022, Crosschapter box 2) such as GWP\* (Allen et al., 2018; Cain et al., 2019) that is based on a minimalistic dynamic model, or CGTP (Collins et al., 2020) that is a relative metric comparing a step change in SLCF emissions with a pulse emission of CO<sub>2</sub>. All these metrics are suited for yearly monitored emissions, e.g. at country level. As sustained emissions are barely encountered in LCA of products, these metrics are left out of the present study to focus on absolute metrics based on pulse emissions.

#### 2.2 Absolute and dynamic metrics






GHGs effective radiative forcing (ERF)\* quantifies the energy gained by the Earth system following an imposed perturbation (Forster et al., 2021). The absolute global warming potential (AGWP) is the integrated ERF. Following Eq. (2), it describes the change in heat flux density caused by a pulse emission, i.e. a Dirac delta function, of a unit mass of gas at  $t_0$ . The AGWP framework can be extended to multi-pulse cumulative ERF calculations,  $\Delta F$ , since product systems can be viewed as a series of pulse emissions and analysed through convolution (Eq. (3), Aamaas et al., 2013):

$$AGWP_i(H) = \int_0^H ERF_i(t) dt = A_i \int_0^H IRF_i(t) dt, \qquad (2)$$

$$\Delta F_i(H) = A_i \int_0^H g_i(t) . IRF_i(H - t) dt , \qquad (3)$$

where H is the time horizon, i the studied gas,  $A_i$  is the radiative efficiency scaling factor in W.m<sup>-2</sup>.kg<sup>-1</sup>,  $g_i$  the temporal emission profile in kg, and  $IRF_i$  is the impulse response function\*.

According to simplified radiative forcing expressions of Etminan et al. (2016), RE of CO<sub>2</sub>, CH<sub>4</sub> and N<sub>2</sub>O depend on CO<sub>2</sub>, CH<sub>4</sub> and N<sub>2</sub>O background concentrations. The same applies for IRF<sub>i</sub>. Decreasing RE<sub>CO2</sub> with increasing CO<sub>2</sub> concentration is partially offset by an increase in climate-carbon cycle feedback (Reisinger et al., 2011) and by CO<sub>2</sub> sink saturation, mainly related to ocean (Raupach et al., 2014). Though, due to current rapid changes in background GHGs concentration and indirect chemical effects complexity, constant RE and IRF<sub>i</sub> over time might be sources of uncertainty for mid- and long-term dCCA. Constant RE and IRF<sub>i</sub> must at least be updated with each new IPCC assessment report. Following the AR6, RE and IRF<sub>non-CO2</sub> values are fixed with 2019-background concentrations (410 ppm CO<sub>2</sub>, 1866 ppb CH<sub>4</sub> and 332 ppb N<sub>2</sub>O). IRF<sub>CO2</sub> is still calculated with 2010-background concentration of 389 ppm CO<sub>2</sub> (IPCC, 2021b; Joos et al., 2013), also similar to AR6.

By contrast, 422 ppm CO<sub>2</sub> was measured on average in September 2024 (NOAA, 2024).

Further down the cause-effect chain of climate change, an additional radiative forcing implies a temperature change. Absolute Global Temperature change Potential (AGTP) is an endpoint metric. It is a well-established method that includes an energy balance climate model (Shine et al., 2005) to compute temperature change after a pulse emission (see Eq. (4)) (Boucher and Reddy, 2008; Fuglestvedt et al., 2010). Applying AGTP with the extended  $\Delta F$  framework defined in Eq. (3) enables to estimate the global-mean temperature change,  $\Delta T$ , to assess multi-pulse scenarios (see Eq. (5)). If the emission profile  $g_i(t)$  is a Heaviside step function, one can note that pulse and sustained metrics become mathematically equivalent.

$$AGTP_i(H) = A_i \int_0^H IRF_i(t).IRF_T(H-t) dt, \qquad (4)$$

$$\Delta T_i(H) = \int_0^H \Delta F_i(t) . IRF_T(H - t) dt , \qquad (5)$$

$$IRF_T(t) = ECS * \sum_{j=1}^{J} \frac{c_j}{d_j} e^{-\frac{t}{d_j}}$$
(6)

- where  $IRF_T(t)$  is the temporally displaced temperature response function of the Earth system. The use of a two-layer energy balance emulator (Geoffroy et al., 2013) enables to simply reproduce the behaviour of a coupled atmosphere—ocean general circulation model. In this simple idealised framework, the heat-uptake temperature is a sum of two contributions (i.e. J = 2 in Eq. (6)): one quick mode representing the planetary surface's response to changes in forcing, and one mode with a much longer relaxation time that takes the large deep ocean inertia into account (Geoffroy et al., 2013).
- AGTP is computed with  $IRF_T$  derived from a constrained ensemble from two emulators: FaIRv1.6.2 and MAGICC7.5.1, both in their AR6 calibration setups. Fast and slow response relaxation times are calculated to match the best-guess assessment of a 3.0°C equilibrium global surface air temperature response to a doubling of atmospheric CO<sub>2</sub> above its preindustrial concentration (Smith et al., 2021). ECS,  $c_j$  and  $d_j$  mean values are given in Tab. 2.
- Analytical resolution of AGWP and AGTP are shown in S2. Compared to AGWP, AGTP increases both the uncertainty and the policy relevance (Levasseur et al., 2016a; Myhre et al., 2013a; Peters et al., 2011b), as it requires an extra step for the climate response but directly gives easy-to-understand temperature changes.

## 2.3 Studied long-lived GHGs features



This study considers some long-lived climate forcers (LLCFs), GHGs whose lifetimes are greater than the time scales for inter-hemispheric mixing (1–2 years) (Szopa et al., 2021). As LLCFs have relatively homogeneous spatial influence in the troposphere, they are considered well mixed, i.e. local emissions impacts can be globally accounted for.

To evaluate the total effect of a GHG, one needs to know its lifetime, its RE and its chemical interaction with other molecules. Components of complex models such as chemical adjustments\* have to be accounted for in emissions-based ERF to provide transparency on climate metrics (Szopa et al., 2021).

## 2.3.1 Carbon dioxide

As the atmospheric CO<sub>2</sub> concentration is governed by a diversity of physical and biogeochemical processes (Joos et al., 2013), IRF<sub>CO2</sub> is usually approximated by a sum of exponentials (see Eq. (7)). Joos et al. (2013) is still the latest multi-model quantification of the response of oceanic and terrestrial carbon sinks to an instantaneous pulse of CO<sub>2</sub> emission (Forster et al., 2021). Coefficients to fit their multi-model mean responses to a pulse emission of 100 GtC are used (see Tab. 2). These coefficients cannot be used to assess impacts on time horizons longer than 1000 years:

$$IRF_{CO2}(t) = \alpha_0 + \sum_{k=1}^{K} \alpha_k e^{-\frac{t}{\tau_k}}, \text{ for } 0 < t < 1000,$$
 (7)

where  $\alpha_k$  represent a CO<sub>2</sub> fraction associated to a nominal timescale  $\tau_k$ , with K = 3, and  $\alpha_0$  is the fraction of emissions that remains permanently in the atmosphere according to this multi-model fit.

#### 2.3.2 Climate-carbon feedbacks

A carbon cycle response happens after the emission of CO<sub>2</sub> and non-CO<sub>2</sub> GHGs: a GHG emission warms the climate, which in turn reduces the carbon sinks uptake efficiency. According to Gasser et al. (2017), Climate–Carbon feedbacks (CCf) are for instance the effect of temperature and precipitation change on net primary productivity and heterotrophic respiration of land ecosystems, or changes in the surface ocean's chemistry. IRF<sub>CO2</sub> from Joos et al. (2013) includes CCf. AR5's attempt to include CCf for non-CO<sub>2</sub> species (Myhre et al., 2013a) was inconsistent (Gasser et al., 2017). AR6 restored consistency by adding CCf to all GHGs after the framework developed by Gasser et al. (2017). Equation (8) indicates the increase in absolute climate metrics ΔAGxx<sub>i</sub> of a gas i due to CCf (Smith et al., 2021):

$$\Delta AGxx_i = \gamma \int_{t=0}^{H} AGTP_i(H-t) \int_{t'=0}^{t} AGxx_{CO2}(t') r_F(t-t') dt' dt, \qquad (8)$$

with  $r_F(t) = \delta(t) - \sum_{i=1}^3 \frac{\beta_i}{\kappa_i} e^{-t/\kappa_i}$  and  $\gamma r_F(t)$  the CO<sub>2</sub> flux perturbation following a unit temperature pulse in kgCO<sub>2</sub> yr<sup>-1</sup> K<sup>-1</sup>.  $r_F$  parameter values are indicated in S3 and an encoded CCf analytical solution is available (see Code availability).

#### 2.3.2 Methane

Oxidation by tropospheric hydroxyl (OH) radical is the major sink of methane followed by other chemical losses – stratospheric and tropospheric halogen losses – and soil uptake (Boucher et al., 2009; Lelieveld et al., 1998; Stevenson et al., 2020). All these sinks lead to a total CH<sub>4</sub> atmospheric lifetime, τ<sub>atm,CH4</sub>, of 9.1 years (Szopa et al., 2021). Methane atmospheric lifetime is shorter than its perturbation lifetime τ<sub>CH4</sub> since an increase in CH<sub>4</sub> emissions decreases tropospheric OH, which in turn enhances its own lifetime and therefore the methane burden (Szopa et al., 2021). Hence a CH<sub>4</sub>-OH feedback factor, *f*, is applied: τ<sub>CH4</sub> = τ<sub>atm,CH4</sub>\*f (see S2). IRF<sub>CH4</sub> is then described with K = 1 and α<sub>0</sub> = 0 (see Eq. (7)).

Methane has a direct radiative effect through absorption of both shortwave and longwave radiation and indirect effects due to its reactivity. CH<sub>4</sub> emissions cause tropospheric ozone production as well as stratospheric water vapour increase (Szopa et

al., 2021). Hence a positive chemical adjustment is attributed to methane and considerably increases the direct effect of CH<sub>4</sub> by a factor of 1.463 (see S2). As in Myhre et al. (2013b), methane influence on aerosols is not included here since these effects have not been confidently quantified to date (Forster et al., 2021). This might change in the future if findings on aerosol-cloud-interaction radiative forcing of O'Connor et al. (2022) are confirmed by future works.

Lastly, oxidation of methane from fossil sources leads to additional fossil  $CO_2$  (Forster et al., 2021). Not all  $CH_4$  oxidises since other sinks as OH radical exist. With a yield of 75%, 1 kg of fossil methane yields the emission of 2.1 kg $CO_2$ , and 1kg of anthropogenic biogenic methane yields to a sink of 0.33 kg atmospheric  $CO_2$  (Boucher et al., 2009; Forster et al., 2021). However, dCCA enables accounting for  $CO_2$  uptake, e.g. put extra negative values to dynamically account for biomass growth or mineral carbonation. Hence, we treat all carbon as equal, namely use fossil methane (Muñoz and Schmidt, 2016), and do not use the biogenic correction to avoid double counting. Equation (9) with no chemical distinction between released carbon from biogenic and fossil sources is then used. One can see that  $CO_2$  is emitted slowly as methane decays, i.e. there is a convolution between IRF<sub>CH4</sub> and AG $_{xx}$  or  $\Delta X_{CO2}$ . The analytical resolution of the convolution is in S4. All these chemical effects significantly impact the radiative forcing of  $CH_4$  (Szopa et al., 2021, Figure 6.12), inducing adapted AG $_{xx}$  formulas.

$$AGWP_{CH4,fossil}(H) = (1+f_1+f_2)A'_{CH4}\tau_{CH4}\left(1-e^{-\frac{H}{\tau_{CH4}}}\right) + Y\frac{M_{CO2}}{M_{CH4}}\frac{1}{\tau_{CH4}^{OH}}\int_0^H e^{-\frac{H-t}{\tau_{CH4}}}AGWP_{CO2}(t)dt + \Delta AGWP_{CH4}(H) \ , \ (9)$$
 where  $f_I$  and  $f_2$  are respectively the ozone and the stratospheric water vapour indirect effects,  $A'$  is the radiative efficiency scaling factor without indirect effects with  $(1+f_I+f_2)A'_{CH4} = A_{CH4}$ ,  $Y$  is the reaction yield from CH<sub>4</sub> to CO<sub>2</sub> molecules,  $M_i$  the molar mass of a gas  $i$ ,  $\tau_{CH4}^{OH}$  the chemical lifetime of methane and  $\Delta AGWP_{CH4}$  is the climate–carbon feedback. AGTP<sub>CH4</sub>,

 $\Delta F_{CH4}$  and  $\Delta T_{CH4}$  are affected the same way. All mentioned climate parameters values are in Table 1 or in S3.

## 2.3.2 Nitrous oxide







Anthropogenic emissions of  $N_2O$  are driven primarily by fertiliser use and the handling of animal waste (Prather et al., 2015). Nitrous oxides loss mainly occurs through photolysis and oxidation by  $O(^1D)$  radicals in the stratosphere, the critical region for  $N_2O$  loss being the tropical middle stratosphere (Canadell, 2021; Prather et al., 2015). The rates of reactions are defined by  $O_3$  and temperature stratospheric vertical profile (Prather et al., 2015). The mean atmospheric lifetime of  $N_2O$  is  $116 \pm 9$  years. A small negative lifetime sensitivity of  $N_2O$  to its own burden leads to an effective residence time perturbation of  $109 \pm 10$  years (Canadell, 2021).  $IRF_{N2O}$  is modelled with K = 1 and  $\alpha_0 = 0$  (see Eq. (7)). The indirect contributions of nitrogen oxides (NO and  $NO_2$ ) push the  $OH/HO_2$  ratio in the other direction than methane through the reaction  $NO+HO_2 \rightarrow NO_2+OH$ , inducing a negative effect on  $CH_4$  lifetime (Stevenson et al., 2020). A positive effect is due to stratospheric ozone depletion (Forster et al., 2021; Szopa et al., 2021). They are relatively minor since they nearly compensate each other.  $A_{N2O}$  is thus scaled with updated value from Forster et al. (2021) so that the AGWP formulae of Eq. (10) evolves from Myhre et al. (2013b):

$$AGWP_{N2O}(H) = A'_{N2O} \left\{ 1 - 1.7 \times (1 + f_1 + f_2) \frac{RE_{CH4}}{RE_{N2O}} + RE_{N2O}^{03} C_f \right\} \times \tau_{N2O} \left( 1 - e^{-\frac{H}{\tau_{N2O}}} \right) + \Delta AGWP_{N2O}(H)$$

$$= A_{N2O} \times \tau_{N2O} \left( 1 - e^{-\frac{H}{\tau_{N2O}}} \right) + \Delta AGWP_{N2O}(H)$$
(10)

where  $A'_{N20}$  and  $A_{N20}$  are radiative efficiency scaling factors in W.m<sup>-2</sup>.kg<sup>-1</sup> respectively without and with indirect effects,  $RE_{N20}^{O3}$  the radiative efficiency through ozone in W.m<sup>-2</sup>.ppb<sup>-1</sup> and C<sub>f</sub> the conversion factor to convert RE from per ppb(N<sub>2</sub>O) to per kgN<sub>2</sub>O. AGTP<sub>N2O</sub>,  $\Delta$ F<sub>N2O</sub> and  $\Delta$ T<sub>N2O</sub> are affected the same way.

Table 1. Climate parameters and associated uncertainties used for simple emission metrics and uncertainty beam calculation.


| Variable            | Definition                                                                                                     | Unit                                 | Value                                                                                                                                                        | Uncer<br>distrib         | Source               |                        |  |
|---------------------|----------------------------------------------------------------------------------------------------------------|--------------------------------------|--------------------------------------------------------------------------------------------------------------------------------------------------------------|--------------------------|----------------------|------------------------|--|
| H                   | Time horizon                                                                                                   | Years                                | 1-1000                                                                                                                                                       | -                        | -                    | (Joos et al., 2013)    |  |
| AGWP <sub>CO2</sub> |                                                                                                                |                                      |                                                                                                                                                              |                          |                      |                        |  |
| Aco2                | Radiative forcing scaling factor                                                                               | W.m <sup>-2</sup> .kg <sup>-1</sup>  | 1.71 x 10 <sup>-15</sup>                                                                                                                                     | 0.21 x 10 <sup>-15</sup> | Normal 1.645σ        | (Forster et al., 2021) |  |
| α0-3                | Coefficient for fraction of atmospheric CO <sub>2</sub> associated with a nominal timescale  Nominal timescale | Unitless                             | $\alpha_0 = 1 - \alpha_1 - \alpha_2 - \alpha_3$<br>$\alpha_1 = 0.2240$<br>$\alpha_2 = 0.2824$<br>$\alpha_3 = 0.2763$<br>$\tau_1 = 394.4$<br>$\tau_2 = 36.54$ | -                        | -                    | (Joos et al., 2013)    |  |
| τ1-3                | Nonmai timescare                                                                                               | Tears                                | $\tau_3 = 4.304$                                                                                                                                             |                          |                      |                        |  |
| AGWPc               | H4                                                                                                             |                                      |                                                                                                                                                              |                          |                      |                        |  |
| A <sub>CH4</sub>    | Radiative forcing scaling factor with indirect effect                                                          | W.m <sup>-2</sup> .kg <sup>-1</sup>  | 2.00 x 10 <sup>-13</sup>                                                                                                                                     | 0.49 x 10 <sup>-13</sup> | Normal 1.645σ        | (Forster et al., 2021) |  |
| тсн4                | Perturbation lifetime                                                                                          | Years                                | 11.8                                                                                                                                                         | 1.8                      | Normal $1.645\sigma$ | (Forster et al., 2021) |  |
| $\tau_{CH4}^{OH}$   | Chemical lifetime                                                                                              | Years                                | 9.7                                                                                                                                                          | 1.1                      | Normal 1σ            | (Szopa et al., 2021)   |  |
| Y                   | Fractional molar yield of CO <sub>2</sub> from CH <sub>4</sub> oxidation                                       | Unitless                             | 0.75                                                                                                                                                         | [0.5 - 1]                | Uniform              | (Forster et al., 2021) |  |
| $AGWP_{N2O}$        |                                                                                                                |                                      |                                                                                                                                                              |                          |                      |                        |  |
| A <sub>N2O</sub>    | Radiative forcing scaling factor with indirect effect                                                          | W.m <sup>-2</sup> .kg <sup>-1</sup>  | 3.6 x 10 <sup>-13</sup>                                                                                                                                      | 1.4 x 10 <sup>-13</sup>  | Normal 1.645σ        | (Forster et al., 2021) |  |
| $	au_{ m N2O}$      | Perturbation lifetime                                                                                          | Years                                | 109                                                                                                                                                          | 10                       | Normal 1.645σ        | Canadell et al. (2021) |  |
| AGTP                |                                                                                                                |                                      |                                                                                                                                                              |                          |                      | (2021)                 |  |
| ECS                 | Equilibrium climate sensitivity                                                                                | K.(W.m <sup>-2</sup> ) <sup>-1</sup> | 0.76                                                                                                                                                         | 0.28                     | Normal 1σ            | (Forster et al., 2021) |  |
| $c_1$               | ECS fractional contribution of the fasterm                                                                     | t -                                  | $c_1 = 0.586$                                                                                                                                                |                          |                      | (Smith et al., 2021)   |  |
| $c_2$               | ECS fractional contribution of the slov<br>term                                                                | v _                                  | $c_2 = 1 - c_1$                                                                                                                                              | see<br>S5                |                      | (Silliul et al., 2021) |  |
| $d_1$               | Fast relaxation time                                                                                           | Years                                | $d_1 = 3.4$                                                                                                                                                  |                          |                      | (Smith et al., 2021)   |  |
| $d_2$               | Slow relaxation time                                                                                           | Years                                | $d_2=285\\$                                                                                                                                                  |                          |                      | (Silliul et al., 2021) |  |

## 3 Sensitivity analysis







In LCAs, climate change CF are often used without related uncertainties, e.g. 1 kgCH<sub>4</sub> = 29.8 kgCO<sub>2</sub>e. Nevertheless, common relative metrics of CH<sub>4</sub> and N<sub>2</sub>O show wide uncertainty ranges: 32%-49% for GWP and 46-83% for GTP (Smith et al., 2021). Olivié and Peters (2013) highlighted that variations in IRF<sub>CO2</sub> and IRF<sub>T</sub> have a considerable impact on common emission metrics, even in linear systems, i.e. for small perturbations. IRF<sub>CO2</sub>'  $\alpha$  and  $\tau$  parameters of Eq. (7) are related to phenomenological modelling, and hence have no physical meaning. They are fitting parameters of a mean that comes from a multi-model analysis. To characterise the IRF<sub>CO2</sub> uncertainty, we randomly use one model's fit coefficients among the 13 ensemble members of Joos et al. (2013) (see S5). In this straightforward and tractable way, we ensure that the constraint  $\alpha_0$  = 1- $\alpha_1$ - $\alpha_2$ - $\alpha_3$  is respected in the probabilistic analysis, but we can't give  $\alpha$  and  $\tau$  specific uncertainty values. Addition of other parameters uncertainties enable us to plot a proper uncertainty range. As done in the AR6, IRF<sub>T</sub>' c and d parameters of Eq. (6) are derived from a constrained ensemble from FaIRv1.6.2 and MAGICC7.5.1, whereas c and d variations are computed from MAGICC7.5.1 ensemble members only (see S6).

Table 2 presents added uncertainties linked to radiative efficiency scaling factors, lifetime perturbations and CH<sub>4</sub> oxidation yield and ECS from Forster et al. (2021). AR6 mostly considers normal uncertainty distribution with [5-95]% confidence range. Monte Carlo simulations (5000 runs for AGWP; 10000 runs for AGTP) are performed to get stable uncertainty ranges. Uncertainties on CCf –  $\gamma$  and on  $r_F(t)$  parameters – are not considered here.

## 4 Results

We first compare in Fig. 1 the dynamic climate change impact of 1kg emission of CO<sub>2</sub>, of CH<sub>4</sub> and of N<sub>2</sub>O. Metric profiles represent responses from present-day (H = 0 year) to the maximum possible long-term time horizon (H<sub>max</sub> = 1000 years if multi-model mean IRF<sub>CO2</sub> from (Joos et al., 2013) is used). AGWP grows up to an asymptotic value, i.e. when GHGs are no longer in the atmosphere. This asymptote comes from AGWP mathematical construct and might lead to bias in long-term interpretations. Differences in orders of magnitude between CO<sub>2</sub>, CH<sub>4</sub> and N<sub>2</sub>O's GWP<sub>100</sub> are well reflected with dynamic AGWP profiles. As for AGTP, it shows a peak temperature change (AGTP<sub>peak</sub>) shortly after emission because of the quick planetary surface response. AGTP<sub>peak</sub> is reached at 10, 6 and 15 years for CO<sub>2</sub>, CH<sub>4</sub> and N<sub>2</sub>O respectively, which fits and extends Ricke and Caldeira (2014)'s observation. Then, a more or less decreasing AGTP is due to deep ocean thermal inertia. Temperature change caused by a CO<sub>2</sub> emission decreases very slowly at long-term H, i.e. CO<sub>2</sub> has a significant long-term impact. Methane causes a notable short-term climate change contribution. It has also a long-term impact from its oxidation into CO<sub>2</sub> emissions: AGWP<sub>CH4</sub> keeps slightly increasing over centuries and AGTP<sub>CH4</sub> does not sharply decrease at long-term perspective. N<sub>2</sub>O behaviour is in-between: GTP<sub>N2O</sub> value begins to decrease with H≈30 years. AGTP temporal emission profiles reflect much more nuances than the use of static GTP values of CO<sub>2</sub>, CH<sub>4</sub> and N<sub>2</sub>O. Table 3 shows CF of both metrics with their associated uncertainties at H routinely reported by IPCC, plus at AGTP<sub>peak</sub>. Slight differences

observed with IPCC values, especially in long-term H, should come from CCf computation: here CCf is analytically resolved whereas numerically resolved by IPCC.

Figure 1. a) AGWP and b) AGTP profiles in logarithmic scale from present-day (H = 0 year) to very long-term perspective (H = 1000 years) after emissions of 1kg CO<sub>2</sub>, 1kg CH<sub>4</sub> and 1kg N<sub>2</sub>O. Uncertainty ranges are computed by varying all parameters listed in Table 1.





Table 2. Characterisation factors of AGWP and AGTP emissions metrics for selected species and time horizons. Uncertainties are calculated with  $1 \times$  standard deviation  $\sigma$ . u% represents the ratio between  $\sigma$  and mean value.

|                 | W.yr.m <sup>-2</sup> .kg <sup>-1</sup> (x10 <sup>-12</sup> ) |                 | u%              | °C.kg <sup>-1</sup> (x10 <sup>-15</sup> ) |                                     |                 | u%              |      |
|-----------------|--------------------------------------------------------------|-----------------|-----------------|-------------------------------------------|-------------------------------------|-----------------|-----------------|------|
|                 | $AGWP_{20}$                                                  | $AGWP_{100}$    | $AGWP_{500}$    | mean                                      | AGTP <sub>peak</sub> (peak timing)  | $AGTP_{50}$     | $AGTP_{100}$    | mean |
| CO <sub>2</sub> | 0.0244<br>± 0.0025                                           | 0.0895 ± 0.0130 | 0.314 ± 0.053   | 14%                                       | $0.54 \pm 0.16  (10  \text{years})$ | $0.43 \pm 0.13$ | $0.39 \pm 0.12$ | 31%  |
| $CH_4$          | $2.01 \pm 0.32$                                              | $2.68 \pm 0.45$ | $3.20 \pm 0.50$ | 16%                                       | $55 \pm 19$ (6 years)               | $5.7 \pm 2.3$   | $3.02 \pm 0.98$ | 36%  |
| $N_2O$          | $6.7 \pm 1.6$                                                | $24.5 \pm 5.9$  | $42 \pm 10$     | 24%                                       | $150 \pm 57$ (15 years)             | $125 \pm 48$    | $93 \pm 35$     | 38%  |

The key aim of metrics is the quantification of the marginal impact of pulse emissions of extra GHG units (Kirschbaum, 2014). Figure 2 compares three pulse emissions being equivalent in terms of  $GWP_{100}$ : 1) an emission of 100 kg of  $CO_2$ ; 2)  $100/GWP_{100\_CH4} = 3.36$  kg of  $CH_4$  emitted; 3) a *mixed\_GHGs* pulse reflecting 2022 global emissions proportion of major GHGs - 99%  $CO_2$ , 0,97%  $CH_4$ , 0,03%  $N_2O$  – (adapted from EDGAR (Crippa et al., 2023)).

Both AGWP and AGTP show that the conversion from CH<sub>4</sub> to CO<sub>2</sub>-equivalent emissions underestimates short-terms impacts and overestimates long-term impacts (Allen et al., 2016). Nevertheless, the notion of CO<sub>2</sub>-equivalent in cases of pulse emission at  $t_0$  makes sense regarding global emissions proportion, especially for mid- and long-term H. Figure 3 shows that CH<sub>4</sub> contribution on temperature change caused by a 1-year emission pulse from all anthropogenic activities is almost equal to CO<sub>2</sub> in the short-term, with a 51:46 CO<sub>2</sub>:CH<sub>4</sub> percentage contribution at H = 10 years. After some decades, temperature change is almost only driven by CO<sub>2</sub> and hence flattens. This is in line with Allen et al. (2022) who support separating SLCFs and LLCFs' contributions in emission targets, especially under short-term perspective. Respectively for  $CO_2$ ,  $CH_4$  and  $mixed\_GHGs$  scenarios, we compute -19%, -68% and -24% between AGTP<sub>200</sub> and AGTP<sub>1000</sub>, which is little

compared to the drop of -40%, -2275% and -163% between AGTP<sub>peak</sub> and AGTP<sub>200</sub>. In a single-pulse framework, AGTP<sub>500</sub> appears to be a representative mean value of this observed temperature change flattening on a long-term perspective. Mean values of these two metrics are presented in Tab. 4.

Figure 2. a) AGWP and b) AGTP profiles for three pulse emissions having the same GWP<sub>100</sub> = 100 kgCO<sub>2</sub>e.




Figure 3. Evolution over time of the relative contribution of  $CO_2$ ,  $CH_4$  and  $N_2O$  to AGTP of 'mean\_mixed\_GHGs' reflecting 2022 global emission (mass ratio: 99%  $CO_2$  - 0,97%  $CH_4$  - 0,03%  $N_2O$ ).

Table 3. Mean AGTP $_{peak}$  and AGTP $_{500}$  associated with the three temporal emission profiles of Fig. 2.b).

|                 | $AGTP_{peak}$            | AGTP <sub>500</sub> |
|-----------------|--------------------------|---------------------|
| $x10^{-14}$     | (°C)                     | (°C)                |
| CO <sub>2</sub> | $5.4 \pm 1.6$ (11 years) | 3.7 ± 1.1           |
| CH <sub>4</sub> | $18.6 \pm 6.4$ (7 years) | $0.41 \pm 0.14$     |
| mixed_GHGs      | $8.2 \pm 1.8$ (8 years)  | $2.85 \pm 0.84$     |
|                 |                          |                     |

Lastly, we address dynamic climate metrics in multi-gas and multi-pulse cases, i.e. what is mostly encountered by assessment communities. This approach is especially interesting to assess long-lasting systems as well as bio-based products that store biogenic carbon while not degraded. To illustrate the potential benefits of using  $\Delta F$  and  $\Delta T$  in such cases, Fig. 4 reflects impacts of two insulating materials with a 50-year lifespan. Expanded polystyrene is fossil-based and straw bale is bio-based. At the production stage *Polystyrene* is more energy intensive while *Straw bale* sequesters much more  $CO_2$  than it emits GHGs. At end-of-life (EoL) stage *Polystyrene* is landfilled and emits very little. *Straw bale* is composted or mulched and releases a large amount of initially captured  $CO_2$  along with  $CH_4$  and  $N_2O$ . Inventories can be found in S7. In a multipulse framework, peak temperature change of product systems might occur decades (see Fig.4), or even centuries after  $t_0$ ,

which significantly shifts the time when temperature change becomes rather stable in a long-term perspective. Hence, instead of  $\Delta T_{500}$ , we propose the metric  $\Delta T_{long-term}$  being 500 years after  $\Delta T_{peak}$  in order to stay representative of the long-term temperature change flattening.


Figure 4. a) Emission profile of  $1\text{m}^2$  of thermal insulators - *Polystyrene* and *Straw bale* – having a thermal resistance of  $7\text{ m}^2$  K W<sup>-1</sup> during 50 years. CO<sub>2</sub>, CH<sub>4</sub> and N<sub>2</sub>O are emitted (see inventories in S7). b)  $\Delta F$  and c)  $\Delta T$  temporal profiles of these two products over 575 years. Dotted grey lines represent H given by IPCC leading to  $\Delta F_{20}$ ,  $\Delta F_{100}$ ,  $\Delta F_{500}$  (b);  $\Delta T_{50}$ ,  $\Delta T_{100}$  (c).

Figure 5. Relative climate change results for *Polystyrene* and *Straw bale* according to common relative metrics – GWP<sub>20</sub>, GWP<sub>100</sub>, GWP<sub>500</sub>, GTP<sub>500</sub>, GTP<sub>500</sub>, GTP<sub>500</sub>, and suggested metrics in this article: ΔF<sub>20</sub>, ΔF<sub>100</sub>, ΔF<sub>500</sub>, ΔT<sub>negative</sub>, ΔT<sub>peak</sub>, ΔT<sub>long-term</sub>. Absolute values are presented in S8. Pulse emissions occurring outside the timeline [t<sub>0</sub>, H] are not considered.

Five main observations can be made: 1) Energy intensive materials contribute to global warming from t<sub>0</sub>. 2) Both dynamic metrics show that temporary carbon storage of bio-based products induces a significant effect in mitigating climate change, at least up to EoL. One can observe a drop in temperature change with a negative minimum, ΔT<sub>negative</sub>, at H = 17 years. Interestingly, negative impacts from temporal carbon storage of bio-based materials last longer with ΔF than with ΔT. 3) ΔT<sub>peak</sub> of *Straw bale* occurs at H = 66 years, i.e. much later than the peak temperature change timing implied by a pulse emission at t<sub>0</sub>. Peak timing as a point of reference for the long-term CF appears relevant. 4) Even if ΔT<sub>peak</sub> of *Polystyrene* and *Straw bale* are similar, *Straw bale* end up with a slightly negative ΔT<sub>long-term</sub>, i.e is the only one fitting with the goal of climate neutrality. 5) Recommended dynamic metrics by IPCC give representative values for short-, mid-, and long-term perspective with ΔF but not for ΔT:

- ΔT<sub>50</sub> fails to capture the most important temperature change contributions of *Polystyrene*. Moreover, it gives a negative value for *Straw bale*, but in a 45-year-lifespan scenario, the result would be the opposite, making the EoL occurrence too sensitive to the H choice.
- unlike suggested by UNEP/SETAC recommendations, temperature change at 100 years is not representative of a long-term impact:  $\Delta T_{100}$  of *Polystyrene* and *Straw bale* are almost equal whereas Fig. 4c and  $\Delta T_{long-term}$  depict a big difference on a longer-term perspective. Indeed, at H = 100 years, *Straw bale* still emits GHGs.

Figure 5 clearly highlights the added value of the dCCA approach: all common relative metrics show positive and rather similar relative impacts between both products. As for recommended dynamic metrics, conclusions vary according to the chosen CF, which might lead to different climate policies, especially about negative and positive impacts of bio-based systems.

#### 5 Discussion




#### 5.1 Towards more clarity on absolute and dynamic climate metrics

Based on Forster et al., (2021) and Smith et al., (2021), this paper summarises main climate metric equations with no hidden parameter values: fast and slow response relaxation time values, as well as ECS value are explicitly given, as are their associated uncertainties;  $CH_4$  and  $N_2O$  indirect effects are explicitly transcribed into metrics equations; CCf analytical solution is calculated and proposed in an open-access code page. Preparation of the seventh Assessment Report of the IPCC (AR7) will begin soon. Each new report is an opportunity to recall climate metric common equations as well as to write down updates in specific gas species metric equations and climate parameters values in a pedagogical manner. The work done in part 2.3, notably inspired by Aamaas et al. (2013) and by what Myhre et al. (2013b) did for  $CH_4$  and  $N_2O$  equations may be supporting materials for this purpose. This would help environmental assessment communities with less expertise and in-depth knowledge of climate models to acquire better comprehension of what underlies absolute metrics and adopt dCCA. Indeed, as mentioned above, dynamic climate metrics are scientifically more accurate and LCA practitioners should use them while assessing bio-based or long-lasting products (lifespan > 5-10 years) (SCORE LCA, 2024).

## 340 **5.2 Dynamic climate metrics interpretation**

## **5.2.1** Emission pulses only at t<sub>0</sub>







AGWP and AGTP can be compared through Fig. 1 and Fig. 2. As these two climate metrics are mathematically different and display different shape types, they are complementary. Radiative forcing metrics are now considered robust and useful (Fuglestvedt et al., 2003). As a time-integrated metric, AGWP temporal profiles keep increasing over centuries when CO₂ is emitted. Contributions of LLCFs stock pollution as well as the effect of CO₂ temporal storage (Zieger et al., 2020) are clearly displayed. Contributions of short-lived well-mixed GHGs are displayed for short H (≤20 years). As AGWP is an integrated metric that does not fluctuate widely, 20-, 100- and 500-year H values appear well suited to get short-, mid- and long-term CFs, although there is no fundamental reason behind these values, and − ignoring common practice − others could be chosen. In agreement with Levasseur et al. (2016a), no hierarchy between H are needed at first sight, H choices being preferably based on where LCA practitioners or climate policies want to put emphasis. Nevertheless, GWP's H can be linked to the discount rate used to evaluate economic damages from each emission (Dhakal et al., 2022). In a cost-benefit framework, internally consistent LCA should therefore prioritise the time horizon that is closest to the discount rate that users of the LCA might then apply. Lastly, AGWP only requires atmospheric response models, needs climate models just when CCf is included, and then embeds less uncertainty than AGTP. Drawbacks of using AGWP are rather linked to the unit − W.yr/m². First, it is not clear for policymakers. Second, calculations are not explicitly linked to ultimate climate-change impacts but to energy imbalance and may not match the expected global surface temperature (Forster et al., 2021; Kirschbaum, 2014).

AGTP is an interesting alternative metric since it directly reflects temperature change. AGTP temporal profiles of all studied gas show a peak temperature because of the rapid planetary surface response. AGTP at mid- and long-term corresponds to the thermal inertia of the deep ocean that maintains the memory of the initial pulse (Shine et al., 2005). Showing these impacts over time is much more refined than giving usual GTP values at 50- and 100-year H. Table 3 shows that the difference between AGTP<sub>50</sub> and AGTP<sub>100</sub> is low and not representative of the difference between AGTP<sub>peak</sub> and AGTP<sub>500</sub>. Such an absolute metric is not frequently used in LCA. Yet, it offers significant advantages. 1) AGTP is able to depict on a same graph emission profiles of both SLCF and LLCF, at least from black carbon ( $\tau = a$  few days) to SF6 ( $\tau =$ 1000 years), even on a 1-year time resolution (Sterner et al., 2014). . 2) Endpoint metrics are most closely aligned with the Paris Agreement and the notion of time of maximum temperature rise (Collins et al., 2020). 3) AGTP is in Kelvin or Celsius degree, a common unit. 4) AGTP<sub>peak</sub> is a curve characteristic that varies with the type of gas and is insensitive to the inadvertent H consensus (Shine, 2009). Picking the peak temperature change is also a form of value judgment. Yet, as the global mean temperature is getting closer and closer to a 2°C peak warming target, knowing when peak temperature occurs relative to t<sub>0</sub> makes this CF particularly relevant. In a systemic approach such as a sustained technological change, dynamic temperature responses would allow identifying the real optimum in terms of temperature increase and its timing. 5) Figure 2 highlights that most human activities emit CO<sub>2</sub>, which implies that most product systems have a characteristic almost asymptotic long-term temperature change impact.

#### 5.2.2 Emission pulses at different timings







Temporally displaced emissions profiles raise an issue about climate change metrics: is the way CFs are designed suited for assessment purposes? Characteristic dCCA inventories with multi-time and multi-gas emission pulses, as shown in Fig. 4, impose thinking differently from using the single gas emission pulse assumed by traditional metrics.

Multi-time and -gas emissions also put emphasis on the benefits of using  $\Delta F$  and  $\Delta T$  for LCA practitioners. To carry out a dCCA, we first propose to get temporal profiles from 0 to 600 years. This gives a visual and detailed comparison between product systems.  $\Delta F$  interpretation is similar to AGWP, i.e. assessments at 20, 100 and 500 years (i.e. short-, mid- and long-term) well accompany the temporal representation. Besides, relative  $\Delta F$  of Eq. (13) computed thanks to dynamic climate change assessment tools (Levasseur et al., 2010; Tiruta-Barna, 2021) might be a way to obtain temporal carbon footprint profiles with a common unit. Here, n is the number of assessed GHGs and i an assessed GHG.

$$\Delta F_{relative}(H) = \frac{\sum_{n} \Delta F_i(H)}{AGWP_{1kg,CO2}(H)},$$
(13)

In a multi-pulse framework, unlike with pulse emission at  $t_0$ ,  $\Delta T_{peak}$  might appear decades after  $t_0$ . In this case, peak timing is a required extra information. Hence, this metric that indicated both peak magnitude and timing occurrence appears even more pertinent. Moreover, when  $CO_2$  is a part of emitted GHGs, which is almost always the case when assessing products and sectors,  $\Delta T_{long-term}$  is a second relevant metric.  $\Delta T_{peak}$  can be interpreted as a flow climate change contribution caused by the rapid temperature response of the Earth after emissions of both SLCFs and LLCFs. It connects well the product system's impact to the global goal of the Paris Agreement to limit the instantaneous peak temperature due in part to the significance of instantaneous elevated temperature in causing heat waves and extreme events (Abernethy and Jackson, 2022).  $\Delta T_{long-term}$  is representative of a stock climate change contribution of a product system, clearly showing that today's emissions will induce a rather stable long-term warming. This recalls that mitigation still leads to global warming, and only "reaching and sustaining net zero global anthropogenic  $CO_2$  emissions and declining net non- $CO_2$  radiative forcing would halt anthropogenic global warming on multi-decadal timescales" (IPCC, 2018), but not reduce it.

These are key features to design transparent absolute and dynamic temperature change CFs. AR6 expresses AGTP and GTP's CF at 50- and 100-year H. For product systems assessment purposes, IPCC is invited to replace AGTP<sub>50</sub> and GTP<sub>50</sub> values by AGTP<sub>peak</sub> and GTP<sub>peak</sub> values in the coming AR7. IPCC could also adopt a long-term temperature change perspective, e.g. AGTP<sub>500</sub> or 500 years after AGTP<sub>peak</sub> occurs, in addition to the common 100-year H. Further research is needed to see the relevance of such metrics to evaluate yearly emissions at a sectoral, national or global level. Lastly, LCA practitioners are encouraged to go beyond CF by implementing a graphical representation that depicts climate impacts on a yearly-basis over centuries. This would enable them to lessen value judgements in assessments.

## **5.3** Uncertainty issues

 $CO_2$  data are less uncertain than  $N_2O$  and  $CH_4$  ones due to low  $CO_2$  radiative forcing scaling factor uncertainty that offset its more uncertain IRF.  $N_2O$  has the highest radiative forcing scaling factor uncertainty. About temperature change metrics, the equilibrium climate sensitivity is known as one of the most uncertain features of the climate system and causes much of the uncertainty in projections of future global warming (Forster et al., 2021; Shine et al., 2005). Indeed, AR6 concludes that there is a 90% or more chance (very likely) that the ECS is between  $2^{\circ}C$  and  $5^{\circ}C$  (Forster et al., 2021). Hence, AGTP's relative uncertainties are about two times higher than AGWP ones (see Table 3). Nevertheless, as ECS explicitly represents a long-term global warming in Celsius degree from doubling  $CO_2$  from pre-industrial level, it also contributes to AGTP and  $\Delta T$  policy relevance. Moreover, the uncertain response time of the climate system depicted by temperature change metrics is a real feature which is not captured by radiative forcing metrics.

As future warming scenarios are not considered, i.e. GHG concentrations are static, uncertainties are even bigger as computed here. How far the potential advantage of AGTP<sub>long-term</sub> to achieve long-term climate targets compared to the disadvantage of being subject to considerably larger uncertainties (Reisinger et al., 2010) is still open to conjecture.

#### 415 **6 Conclusion**






While we are becoming more and more aware of the Earth's climate system's complex functioning, it is critical to keep clear and understandable climate metrics for the LCA community. It might indeed be difficult to make connections between the complexity of climate models and successive recommendations as and when IPCC reports are presented. As preparation for the next IPCC assessment will begin soon, this paper highlights the importance to clearly recall dynamic equations that underlie climate metrics and to properly gather updated climate parameter values with associated uncertainties. The overview of up-to-date climate data has been presented here with this pedagogical purpose in order to help environmental assessment communities adopt consistent dynamic climate metrics.

Absolute and dynamic metrics enable us to properly represent specific behaviours of different climate forcers over time. There is a growing interest in using them to take the analysis one step further than with  $CO_2$ -relative and static metrics (GWP, GTP). But while climate metrics are designed with single gas pulses emitted at time zero, LCA modelling of products and systems generally leads to multi-pulse with multi-gas emission profiles. Hence, usually recommended and used CFs might not be suitable for dCCA. To investigate that, we have compared main dynamic metrics: AGWP and AGTP for one-pulse emissions, and their multi-pulse emissions equivalent,  $\Delta F$  and  $\Delta T$ . Cumulative radiative forcing and temperature change metrics appear to be complementary. Radiative forcing metrics are quite simple to compute and give less uncertain results. With impacts that keep growing with time, they display in a more pronounced manner the impact of very long-lasting  $CO_2$  and temporary carbon storage. Global temperature change endpoint metrics are more complex and uncertain, but meet both scientific completeness and pragmatic policy choice. First, they represent climate impacts in the common Celsius degree unit, that more strongly resonate with the global target of halting and maintaining global warming below  $2^{\circ}C$  above

the preindustrial level. Second, the graphical representation captures well the nature of both LLCFs and SLCFs, i.e. integrates both a short-term perspective with associated target overshoot concerns and temperature change that will remain for centuries. Lastly, CFs proposed here ( $\Delta T_{peak}$ ,  $\Delta T_{long-term}$ ) are an attempt to get rid of the time horizon issue that has plagued the LCA community for so long.

Assessments using CO<sub>2</sub>-equivalent climate impacts give sufficiently reliable results to go towards mitigation. Nevertheless, to achieve ambitious objectives such as climate neutral product systems, this work showed that climate policy should gain in consistency by adopting temporal metric profiles and selecting some specific values in addition or in substitution to relative metrics. Hence, environmental assessors are invited to display dynamic assessment results up to 600 years after  $t_0$  and to adopt  $\Delta F_{20}$ ,  $\Delta F_{100}$ ,  $\Delta F_{500}$ ,  $\Delta T_{negative}$ ,  $\Delta T_{peak}$ ,  $\Delta T_{long-term}$  as new climate change CFs. These metrics particularly aim to reflect whether or not climate neutrality of product systems is achieved, either at short-, mid- or long-term perspectives according to the policy objectives for which the chosen metrics are applied. IPCC could support this by adopting AGTP<sub>peak</sub> and AGTP<sub>500</sub> (or directly AGTP<sub>long-term</sub>), especially if their relevance to evaluate climate policies at national and global scales are confirmed by future research works.

#### Appendix A: Glossary






<u>Characterisation factor (CF):</u> produced by modelling consequences of withdrawals and discharges on ecosystems, human health or on resources, a characterisation factor provides the contribution of an elementary flow to an impact category. For climate metrics, a CF converts the impact of 1 kg of a GHG emission or uptake into a defined unit and time horizon.

<u>Chemical adjustments:</u> the perturbation of a single emitted compound is not limited to its direct radiative forcing, but can induce subsequent chemical reactions and affect the abundances of other climate forcers. As an example, CH<sub>4</sub> emissions cause tropospheric ozone production as well as stratospheric water vapour increase.

Dynamic climate metrics: absolute metrics used in a temporal dynamic approach that considers a multi-pulse framework with storages and emissions timing. First attempts on dynamic climate change assessment calculated the benefits implied by a delayed emission, but still with a fixed time horizon (Fearnside et al., 2000). Levasseur and her co-workers (2010) extended the approach by calculating absolute and relative radiative forcing metrics on a yearly basis over several hundreds of years. Here, relative metrics are not included in this terminology. This paper addresses in particular two dynamic climate metrics: cumulative radiative forcing ( $\Delta F$ ) and global mean temperature change ( $\Delta T$ ).

Effective radiative forcing (ERF): ERF is employed as the central definition of radiative forcing in AR6. It quantifies change in net downward radiative flux at the top-of-atmosphere following adjustments in both tropospheric and stratospheric temperatures, water vapour, clouds, and some surface properties (Forster et al., 2021). Hence, AR6 includes tropospheric rapid adjustments (+5% for CO<sub>2</sub>, -14% for CH<sub>4</sub> and +7 % for N<sub>2</sub>O) to the stratospheric-temperature-adjusted radiative forcing equations of Meinshausen et al. (2020) to get ERF and RE values (Smith et al., 2021).

Impulse response function (IRF): describes the atmospheric decay of an emitted species. Its general formulation follows one or several exponential decay functions (Joos et al., 2013),  $e^{\frac{-t}{\tau_i}}$ , where  $\tau_i$  is the e-folding time representing the perturbation lifetime of a gas i

Radiative efficiency (RE): equals the ERF for a change in the atmospheric abundance. It is converted from W.m<sup>-2</sup>.ppb<sup>-1</sup> to W.m<sup>-2</sup>.kg<sup>-1</sup>by multiplying with  $(M_A/M_i)*(10^9/m_{atm})$ , where  $M_A$  and  $M_i$  are the molecular weight of dry air (28.97 g.mol<sup>-1</sup>) and the studied gas *i* respectively, and  $m_{atm}$  is the mean dry mass of the atmosphere (5.1352 x  $10^{18}$  kg) (Myhre et al., 2013b).

Short-lived climate forcer (SLCF): SLCFs include aerosols and chemically reactive gases, both affecting climate (Szopa et al., 2021). Depending on the species, their perturbation lifetimes,  $\tau$ , range from a few hours to more than a decade. If  $\tau$ <1-2 years, SLCFs are also called near-term climate forcers (NTCFs) i.e. are spatially highly heterogeneous. Otherwise they might also be called short-lived well-mixed GHGs.

#### 475 Code availability



The dynamic climate change assessment tool is accessible here: https://gitcdr.univ-ubs.fr/DynCC/Metrics\_assesment\_tool

#### **Author contribution**

**V.Z.:** Writing – original draft preparation, Conceptualization, Methodology, Visualization. **T.L.:** Writing – review & editing, Conceptualization, Methodology, Supervision, Funding acquisition. **S.G.:** Writing – review & editing, Conceptualization, Supervision. **Y.G.:** Software. **M.B.:** Writing – review & editing, Supervision. **T.G.:** Writing – review & editing, Resources, Validation. **Y.H.:** Software.

# **Competing interests**

The authors declare that they have no conflict of interest.

## Acknowledgments

The authors would like to acknowledge the financial support of the LOCABATI project funded by the French Agency for Ecological Transition (ADEME), as well as of the Centre Scientifique et Technique du Bâtiment (CSTB). We thank Serguei Sokol for interesting inputs on uncertainty issues, as well as William Collins, Fortunat Joos and Zebedee Nicholls for sharing their datasets and advice.

#### Nomenclature

AGTP Absolute global temperature change potential **AGWP** Absolute global warming potential AR6/AR7 Sixth / Seventh assessment report CCf Climate-carbon feedback CF Characterisation factor CH<sub>4</sub> Methane CO<sub>2</sub>Carbon dioxide CO<sub>2</sub>e Carbon dioxide equivalent Climate-carbon feedback contribution to AGTP ΔAGTP  $\Delta$ AGWP Climate-carbon feedback contribution to AGWP ΔF Cumulative radiative forcing in a multi-pulse framework  $\Delta T$ Global mean temperature change in a multi-pulse framework dLCA Dynamic life cycle analysis **EoL** End-of-life **ERF** Effective radiative forcing **GHG** Greenhouse gas **GTP** Global temperature change potential **GWP** Global warming potential Η Horizon time **IPCC** International panel on climate change  $IRF_{i}$ Impulse response function describing the atmospheric decay of a gas i  $IRF_T$ temporally displaced temperature response function of the Earth system LCA Life cycle analysis **LCIA** Life cycle Impact Assessment LLCF Long-lived climate forcer **MIP** Model intercomparison project  $N_2O$ Nitrous oxide ОН Hydroxyl radical RE Radiative efficiency RF Instantaneous radiative forcing **SLCF** Short-lived climate forcer

σ

Normal standard deviation

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
