# Peer review of "Climate change metrics: bridging IPCC AR6 updates and dynamic life cycle assessments"

_EGUsphere, 2025_

## Author Response (AR1)

**1st reviewer**

Dear referee,

Thanks for your meaningful response. As you mentioned, it is a cross-disciplinary article addressed to two communities: Climate science and LCA. After your review, the importance of being published by an Editor that focuses on climate science rather than on LCA appears more clearly. Indeed, climate science is the foundation of climate metrics and IPCC has such an influence for the LCA community on metrics and characterization factors.

Your comments also highlight the benefits of such cross-disciplinary discussions and contribute to the global aim of the article: raising climate scientist's awareness on their influence and on LCA case studies that might induce the need of other indicators, as well as helping the LCA community on climate change metrics understanding.

We want to mention that the first part of this article is not just a review. Dynamic characterization curves for  $CO_2$ ,  $CH_4$  and  $N_2O$  from 0 to 1000 years with climate-carbon feedback analogically resolved are new. Unlike in IPCC AR6 Chapter 7, fast and slow response relaxation time values, as well as ECS value are here explicitly given, as are their associated uncertainties. Overall, we have put efforts to display proper dynamic metrics uncertainties, which is highly relevant knowing that climate change metrics uncertainties are almost never discussed by the LCA community. Lastly, we do believe that absolute and dynamic metrics applied to multi-timing pulses we present and address here still need further research works and open discussions.

Lastly, we particularly thank you for your major comments that really helped to correct some statements or to make them clearer.
* * *
Referee comments are in italic. Author answers are just below

10: "do not pay attention" – this is a sweeping statement, and I don't know the extent to which it is true.

We do agree, this statement is proposed to be tempered: "complex and diverse climate models are generally not deeply investigated by Life Cycle Assessment communities"

10: "last IPCC report" – this is vague. I think you mean the most recent AR6 WG1 report

We agree with both referee and editor comment and propose: "Sixth Assessment Report of the Intergovernmental Panel on Climate Change"

21: More later on this but AGTP\_long-term is meaningless at this point in the paper.

We agree and propose "peak and long-term temperature change"

35: I don't understand the difference between CF and an absolute metric such as AGTP or AGWP. Is there a difference?

Here metric describe the equation, i.e. AGTP or AGWP. CF is the value at a specific time for a specific gas mass. CF is a typical LCA term to characterize a pollutant with a value.

\*45: "remains static and relative to CO2": I do not understand this. In its initial presentation it was both AGTP and GTP, and even in IPCC WG1 AR6, the AGTP is presented. I also don't accept that it remains static. There have been multiple uses of a dynamic version of the GTP, which depends on the time to a given target – for example Figure 8.30 of IPCC WG1 AR5 report and references therein. I assume that this is what "dynamic" means in this paper.

We totally agree and propose to adapt this to the LCA community. Indeed, dynamic relative metrics sometimes discussed by climate scientists are never used by the environmental assessment communities.

"it [GTP] remains relative to CO2 and is not dynamically used by the LCA community."

94: Table 1 is barely referred to in the text. Is it necessary?

We agree that it is not necessary in the manuscript. As it highlights the evolution of climate parameters over IPCC reports, which is interesting to see limits of dCCA, Table 1 is proposed to move in the supplementary material.

\*130: The AGTP has be presented in both pulse and sustained forms, so this generalisation of the AGTP as "pulse" metric is incorrect. There is also a literature that shows the close relationship between the sustained AGTP and pulse GWP and I was surprised that this point was never made, as I think it is useful information for the target community. This equivalence underpins the GWP\* approach in the various Allen et al. papers cited here.

We agree and propose to better clarify. Indeed, it was stated that the present paper focuses on absolute metrics based on pulse emissions, as sustained emissions are barely encountered in LCA of products.

Line 95: "Integrated or sustained temperature change metrics (iAGTP, iGTP, SAGTP, SGTP) [...] are not considered here since they have similar behaviours as integrated radiative forcing metrics (AGWP,  $\Delta$ F, GWP), at least for long-lived climate forcers (Azar and Johansson, 2012; Collins et al., 2020; Levasseur et al., 2016a)".

Line 130 : "If the emission profile  $g_i(t)$  is a Heaviside step function, one can note that pulse and sustained metrics become mathematically equivalent."

150: "homogeneous climate influence" – maybe there is some confusion between the homogeneity of the forcing agent and the homogeneity of the climate response which is generally characterised by marked land-ocean differences and Arctic amplification.

We agree and propose: "homogeneous spatial influence" to focus on the forcing agent homogeneity.

185: I am not sure where this 1.463 comes from. In Forster et al. (2021) the indirect effects of methane are presented in a different manner (in a radiative efficiency formulation) rather than as some fixed multiplicative factor applied to the direct methane radiative efficiency (see their Section 7.6.1.3).

Climate equation from Myhre et al. (2013b) (see 8.SM.11.3.2) is taken. Following Foster et al. (2021) updates, f1 = 1.4/3.89 = 0.36; f2 = 0.4/3.89 = 0.103. Thus, we increase the direct effect of CH4 by 1 + f1 + f2 = 1.463.

220: Table 2 repeats material that is also in Table S2.

We agree and propose to delete rows of Table S2 that already are in Table 2.

256: Many of these numbers differ a bit from those in Forster et al. (2021). The reason for this should be explained.

We propose to add the following explanation: "Slight differences observed with IPCC values, especially in long-term H, should come from CCf computation: here CCf is analytically resolved whereas numerically resolved by IPCC."

262: Why 3.36 kg? I think I know why, but if this paper is meant to serve a pedagogical purpose, it should be made explicit.

\*274: Given that the time of peak warming varies little between gases, I can't see that AGTP\_longterm serves a usefully different purpose to AGTP(500) and I am not even certain that AGTP\_peak is that different to say AGTP(15) – what is presented feels an unnecessary complication. I was not really so convinced about how either of these extra metrics would make a decisive difference to decision making.

There's some confusion in the reviewer's comment between peak warming timing from a single gas emission at t0 (gas specific) and occurrence of pulse emissions (life cycle inventory specific). For instance, Figure 4 shows that peak temperature change can appear at 62 years. Product systems that have a peak > 100 years after do t0 exist. We propose to add:

"Timing correlation between AGTPpeak and AGTPlong-term appears relevant as peak temperature change of product systems might occur decades (see Fig.4), or even centuries after t0, which significantly shifts the time when temperature change becomes rather stable in a long-term perspective."

282: Maybe "expanded" rather than "expensed" is a better word?

Corrected.

298: Maybe I miss it, but I do not think IPCC recommends particular values of H. It would be better to say they present results for particular values of H.

We agree and propose: "Dotted grey lines represent H given by IPCC"

307: "longer benefits" – I don't know what this means. It may be "longer perceived benefit"? Benefits do not depend on the chosen metric.

We agree and propose to be clearer: "Interestingly, negative impacts from temporal carbon storage of bio-based materials last longer with  $\Delta F$  than with  $\Delta T$ ."

308: "much later ..." I do not understand this. The GTP does not show a peak at t\_o

We agree to clarify this statement and propose: "much later than the peak temperature change timing implied by a pule emission at t0."

\*324: "lacks of clearly". I think this is too negative. Between Smith et al. (2021) and Forster et al. (2021) I have only identified one thing that seems clearly missing to recalculate the metrics and that is the two-term climate response formulation that is used to calculate the AGTP (but that can be found quite easily on the Chapter github page Chapter-7/notebooks/335\_chapter7\_generate\_metrics.ipynb at main · IPCC-WG1/Chapter-7 · GitHub)). I think this "blanket" criticism of the IPCC chapter is unfair on the IPCC authors. This paper should be specific about what is missing, rather than making general negative statements.

We agree to be more specific and propose:

"Supplementary material of AR6 WG1 chapter 7 (Smith et al., 2021) summarises main climate metric equations, but has some limitations: fast and slow response relaxation time values, as well as ECS value are not explicitly given, as are their associated uncertainties;  $CH_4$  and  $N_2O$  metrics equations with their indirect effects are not recalled; CCf analytical solution is not calculated in the report nor in an open-access code page."

343: "not clear for policymakers" – in my experience, policymakers are more interested in, and understand well, metrics which are presented relative to CO2.

We agree, and think that °C is understandable as well.

360: "vast majority of human activities emit CO2" Consider rewording.

We propose: "most human activities emit CO2".

386: 500 years after AGTP peak is approximately 500 years (give or take a few years).

Not necessarily, notably for long-lasting bio-based products (see response relative to line 274). Nevertheless, in most cases, temperature change has already flatten 500 years after production. Hence propose to let this open:

"IPCC could also adopt a long-term temperature change perspective, e.g. AGTP500 or 500 years after AGTPpeak occurs"

\*395-396: "ECS uncertainty is not a barrier ..." I looked back at the paper referred to here, and the uncertainty is not a barrier for RELATIVE metrics, as CO2 is affected by ECS uncertainty in a quite similar way to other gases. But since this paper focuses on absolute metrics, this statement is misleading here

We do agree. We propose to remove the sentence in order to just have this paradox: more uncertainty on one side and more understanding on the other side.

"AGTP's relative uncertainties are about two times higher than AGWP ones (see Table 3). Nevertheless, as ECS explicitly represents a long-term global warming in Celsius degree from doubling  $CO_2$  from pre-industrial level, it also contributes to AGTP and  $\Delta T$  policy relevance. Moreover, the uncertain response time of the climate system depicted by temperature change metrics is a real feature which is not captured by radiative forcing metrics."

398: "uncertain response time" – I am aware that the value of ECS impacts on the response time of the climate system, but is this what is meant here?

Not really. We propose to clarify (see just above).

423-424: dT\_negative is only introduced briefly in the main paper. For an individual forcing agent, does this differ from dT\_peak as it just reflects either a negative forcing agent or the removal of a positive forcing agent? In the example presented it seems to arise from a particular combination of removals and emissions. Won't it depend on that particular combination in any application? As noted above, maybe the most relevant time horizon for a temperature related metric is the time at which some target is likely to be exceeded

For an individual forcing agent, dT\_negative is just the opposite of dT\_peak. Hence, we propose to remove dT\_negative from the proposed CFs , and keep the recommendation to use this indicator only for environmental assessors.

**Added reference:**

Azar, C. and Johansson, D. J. A.: On the relationship between metrics to compare greenhouse gases - the case of IGTP, GWP and SGTP, Earth System Dynamics, 3, 139–147, https://doi.org/10.5194/esd-3-139-2012, 2012.

**2nd reviewer**

Dear referee,

Thanks for your exhaustive response. The fact that you found the article useful and concerning at the same time highlights the benefits of such cross-disciplinary discussions provided with this article. Most of your comments really help this paper to gain in consistency and clarity. Others, we believe, are part of a debate we would be glad to have once the paper is published.

We particularly thank you for having pointed out the lack of clarity between proposed metrics and policy objectives. The Paris Agreement notably states to develop capacity-building strategies to mitigate and adapt at the national, subnational and local levels. And as absolute metrics are valid to assess small perturbations about current concentrations, we propose to explicitly restrict the objectives of our current work to product systems studies and their comparisons according to physical science.

Firstly, proposed metrics and graphical representation of impacts have the advantages to depict short, mid-, and long-term perspectives. This enables the end user to set a time horizon only if needed. We propose to make this clearer. Indeed, we do think that policy objectives have to be adopted first. And then, as stated by the IPCC emission metric definition, comes the choice of metrics that aim to tell whether objectives are about to be reached or not.

Secondly, existing global policy frameworks are already declined in many near/net-zero targets by 2050 sector by sector, at national or at company levels. As an example, the French environmental regulation on new residential building construction has set specific climate metrics to help mitigate embodied emissions of the sector. With our proposal, a product system that shows negative peak temperature change and a near-zero long-term temperature change are in line with local declination of global climate objectives. Here temperature change metrics fulfil the reporting regarding one sector policy objectives.

Thirdly, after your comments, the relevance of proposed metrics to evaluate yearly emissions at a sectoral, national or global level are clear perspectives for future research work. We also noted your comments on the terms 'climate neutrality' and 'carbon neutrality' and we will just use the former one.

To conclude, we agree that absolute and dynamic metrics add complexity for LCA practitioners to get GHGs inventories and provide results. We provide an example of a tool that can be implemented and further develop to ease dynamic assessments. We also observe that there is willingness to move forward (see ref. Score LCA (2024)) in this respect. We do believe that once the results are obtained, it is not more difficult to interpret and take appropriate policy actions. Evidence of this is for instance the impressive adaptation of the all French building sector after the adoption of a brand-new 'semi-static, semi-dynamic' climate indicator in French environmental regulation on new buildings. After yout review, it appears clear to us that a publication of this article in a climate science oriented journal will help different scientific communities to move forward.
* * *
Referee comments are in italic. Author answers are just below

L16: I find no discussion in the paper why time horizons of 20, 100 and 500 years for cumulative radiative forcing are 'sufficient' – largely because the paper does not discuss 'sufficient for what'.

We agree and propose clearer sentences:

line 339-340: "As AGWP is an integrated metric that does not fluctuate widely, 20-, 100- and 500-year H values appear well suited to get short-, mid- and long-term CFs, although there is no fundamental reason behind these values, and – ignoring common practice – others could be chosen"

line 428-429 : "to adopt  $\Delta F_{20}$ ,  $\Delta F_{100}$ ,  $\Delta F_{500}$ ,  $\Delta T_{negative}$ ,  $\Delta T_{peak}$ ,  $\Delta T_{long-term}$  as new climate change CFs to see short-, midand long-term mitigation potential that should fit any mitigation policies."

L18: positive and negative peaks do not alleviate the time horizon issue, they simply pick other time horizons — ones motivated by physical science rather than policy objectives. That might make them more relevant from a physical science perspective but almost by definition less relevant for policy (unless and until they are then used, as a second step, to relate to policy objectives, which the authors do not discuss).

We understand your point and partly agree with you. We propose to clarify in the rest of the paper that proposed metrics are especially appropriate for product system policy objectives. Regarding product systems, peak temperature change might happen for instance at 11 years after production for one solution and at 145 years after production for another. A fixed 100-time-horizon has almost no chance to capture this. Furthermore, we do think that being driven by physical science can also be a political choice.

We propose: "Positive and negative peaks, as well as long-term temperature change, partly alleviate the time horizon decision issue while assessing product systems"

**L22: no justification given for 600 years**

We agree and propose to make clearer that a timeline of 600 years for graphical impact representations is just a proposition to see on one side both short- and long-term cumulative radiative forcing impacts, and on the other side, short- or mid-term peak temperature change as well as sort of stabilized long-term temperature change. Even if it is stringent to focus mitigation policies for the 21st century, it is also relevant to see if proposed actions do not just postpone climate impacts for future generations.

"environmental assessors are invited to display dynamic assessment results up to 600 years after to."

L40: "poor indicator of net-zero timing": what does that mean? If net-zero is defined based on GWP100 (as it is in most countries' policy documents), then GWP100 is a perfect indicator of net-zero timing. If the authors propose a different definition of 'net-zero', they need to introduce the different definition and explain why it is better. However, I suspect this would go well outside the scope of this paper.

We agree that this statement is an over-simplification of Fuglestvedt et al. (2018) and prefer to remove the statement.

L44: GTP is more relevant only if the policy objective is cost-effective abatement relative to peak global temperature, but not if the policy objective is cost-benefit based abatement (as per Tol et al 2012, and updated in the most recent IPCC WGIII assessment). GTP is not more policy relevant simply because it is further down the cause-effect chain. Same applies to L145-147.

We understand your point and remove the statement:

"Global Temperature change Potential (GTP) is explicitly linked to temperature change (Shine et al., 2005), but remains relative to CO2"

L52: the claim that these metrics 'incorrectly' assess the impact of SLCFs is not substantiated – it depends entirely on what the metric is intended to achieve (which, in turn, ought to depend on the policy objective).

The aim of the UNEP/SETAC Life Cycle Initiative is to provide insights to better reflect impacts of any climate forcers (both short- and long-lived). And according to Allen et al., (2018), the chosen metrics do not fully fulfil this objective. We propose to clarify: "these CO2-equivalent metrics have a poor temporal correspondence with

temperature responses for short-lived climate forcers\* (SLCFs), i.e. are a poor indicator of temperature stabilisation (Allen et al., 2018b)."

Sterner et al, (2014) and Lund et al., (2020) latter cited in the paper showed that AGTP is well suited to simultaneously depict short- and long-lived climate forcers.

L58: the term 'dynamic metric' is not defined. I'm thinking of dynamic GTP as a dynamic metric, but that is not what the authors have in mind as far as I can tell.

The "\*" to refer to the glossary has been forgotten. Definition in Appendix A – Glossary explains that dynamic refers here only to absolute metrics. For more clarity, we propose to add in the definition "Dynamic GWP or GTP are not included in this terminology."

L60: the authors should discuss whether/why one can't simply apply pulse-emission metrics to emissions and removals occurring at different times

We agree and propose to complete the existing sentence: "the way most climate metrics and CFs are designed, i.e. based on single gas emission at time zero (t0) or on aggregated emissions and removals into one CO2-equivalent pulse at t0"

L75: "to design strong sustainability" – very unclear what this means and the article does not provide any analysis that demonstrates how it would measure achievement of this goal

We agree and propose: "to discuss to what extend  $\Delta F$  and  $\Delta T$ , two absolute and dynamic metrics, better reflect mitigation policies"

L105: it would have been helpful for Table 1 to clarify how much of the change in metric values is due to reevaluation of indirect effects of methane vs changing baseline concentrations and radiative efficacy.

We prefer not to focus on that (this is a great topic for another paper, or maybe IPCC reports should systematically show this), and propose to move Table 1 in the Supplementary Material, just to show that dynamic climate change assessment is based on static parameters that need at least to be updated after every IPCC assessment report release.

L110-147: I'm missing a discussion of the indirect effects of methane and the large uncertainties this brings, including temperature feedbacks on those effects that result in different fatness of the warming tail for methane

While we agree with the reviewer, indirect effects on climate from methane and temperature dependence of those effects are indeed out of the scope of this study.

L182-187: this discussion is too short in my view, given that a large part of the changes in GWP100 estimates from IPCC AR2 to AR6 has been due to revisions of indirect effects; in addition, whether one includes the temperature dependence of those effects has a material bearing on the fatness of the warming tail of methane.

Again, while we agree with the reviewer, we feel this is not in the scope of our paper. "This might change in the future if findings on aerosol-cloud-interaction radiative forcing of O'Connor et al. (2022) are confirmed by future works. "Appears sufficient to us to let climate scientists include and discuss wider indirect effects in future works."

L188-193: this discussion misrepresents the reason why there is a distinction between biogenic and fossil CO2. Muñoz and Schmidt discuss the fossil CO2 component that may be accounted for separately, but that would argue for using only biogenic CH4 rather than only fossil CH4. More importantly, biogenic CH4 assumes a removal of CO2 during its production (e.g. by growing grass that is then consumed by ruminants or ends up in landfills); this CO2 removal is normally part of the short-rotation carbon cycle that does not normally get counted in dCCA as far as I know. Therefore it would mischaracterise the impact of a biogenic CH4 emissions if the fossil CH4 warming

were applied. The distinction between fossil and biogenic CH4 needs to be explained more clearly so that users can make a distinction where this is appropriate (while noting that the difference is not large in general).

CO2 removal is counted in every proper dCCA dealing with bio-based materials (Duval-Dachary et al., 2024; Levasseur et al., 2012; Pittau et al., 2018; Zieger et al., 2020). We propose slight modifications to better use Muñoz and Schmidt work:

"dCCA enables accounting for  $CO_2$  uptake, e.g. put extra negative values to dynamically account for biomass growth or mineral carbonation. Hence, we treat all carbon as equal, namely use fossil methane (Muñoz and Schmidt, 2016), and do not use the biogenic correction to avoid double counting."

L264: odd statement; of course GWP100 is unitless; it needs to be multiplied with the native-unit non-CO2 emissions to obtain CO2-equivalent emissions. The confusion sometimes arises in LCA that "Global Warming Potential" is also the name of the overall impact factor, but GWP100 is indeed and always unitless.

Another reviewer proposed to point out the possible confusion. Indeed, it can also be stated that GWP100 is not unitless: it is in e.g. kgCO2 per kgCH4. As it moves the discussion outside the scope of the paper, we agree to remove this statement.

L301: it would be helpful to explain how emissions that occur at different times are represented when the 'standard' pulse emission metrics are used. Are they assumed to occur concurrently and hence simply added up? (This makes little sense since the period covered by the time horizon H would be different). Some more explanation and discussion is needed.

Standard relative metrics aggregate emissions, i.e. pulse emissions occur concurrently. We propose to add:

"Pulse emissions occurring outside the timeline [t0, H] are not taken into account."

L310-317: the discussion is useful in that it highlights that point-year metrics are highly variable with the time horizon and thus do not capture the complete picture; but the discussion completely misses the point that point-year metrics only ever make sense if the point-year has a particular policy relevance. Using a different time horizon determined by whenever warming peaks from an emission might describe interesting physical aspects but this does not ensure greater policy relevance (and it remains patchy in describing the complete picture).

We do think that this article is useful for publication exactly to have a broader discussion on such points.

We agree that it remains patchy in describing the complete curve by stating that it is another form of value judgement. We do not agree to fit a metric with a policy agenda that do not match GHG lifespan, for instance. The Paris agreement global objective is stringent mitigation in the short-term to reach and maintain near-zero emissions in the mid- and long-term. But if we set for instance 2050 as a policy relevant point year, this year  $\Delta T_{25}$  needs to be assessed, next year  $\Delta T_{24}$  ... which makes little sense from a LCA practitioner's perspective.

Hence, we do state that absolute metrics give absolute results with no explicit reference to any remaining stock or global temperature target. Policy relevance is obtained when choices toward nearly climate neutral product systems are taken according a time perspective set by the policy itself.

L341: the claim that there is 'no reason to prioritise a specific time horizon' is inconsistent with the fact that the time horizon for GWP can be linked to the discount rate (see IPCC WGIII chapter 2). Internally consistent LCA therefore should prioritise the time horizon that is closest to the discount rate that users of the LCA might then apply - not all time horizons are equal.

Initially the idea behind was: there is no reason to prioritize CH4 mitigation (small perturbation lifetime, high radiative efficiency) over CO2 mitigation (very long perturbation lifetime, small radiative efficiency). Both needs to be drastically mitigated.

We agree to add your point and propose:

"In agreement with Levasseur et al. (2016a), no hierarchy between H are needed at first sight, H choices being preferably based on where LCA practitioners or climate policies want to put emphasis. Nevertheless, GWP's H can be linked to the discount rate used to evaluate economic damages from each emission (Dhakal et al., 2022). In a cost-benefit framework, internally consistent LCA should therefore prioritise the time horizon that is closest to the discount rate that users of the LCA might then apply."

L352: the claim that Kelvin is a unit that everybody understands and therefore AGTP peak is highly relevant is too simplistic. If an LCA tells a consumer that a kg of meat contributes 3.7 pico Kelvin to global warming, what will they do with this 'easy to understand' information?

We totally understand your point. The main point is that it is more understandable than pico W.yr/m². More profoundly, neither nano Celsius degree nor kilogram CO2 equivalent appears more understandable to link local emissions and global targets. But as stated in the article, Celsius degree is a common unit and endpoint metrics are more relevant to address both temperature change and timing of occurrence (e.g. the time of maximum temperature rise).

We propose lower down the positive point: "AGTP is in Kelvin or Celsius degree, a common unit"

L357: "getting close and closer to the 2°C warming target" – please note there is no 2°C 'warming target' in the Paris Agreement, the target is to limit warming to well below 2°C and to purse efforts to limit warming to 1.5°C

We agree and therefore did not add Paris Agreement in this statement. The already very challenging 2°C warming target is largely used in the climate science literature. We propose to change «the» by «a».

L356: the authors miss the point that GTP makes sense as metric only if there is an externally imposed time horizon (namely the time when global temperature is expected to peak). How will users of dCCA (or LCA) interpret LCA results that use different AGTP peak time horizons for different gases (e.g. when comparing the climate impact from different processes that emit a different mix of gases with different time horizons) – what sense do they make of this information and how will it lead to better policy decisions?

The point of dCCA and dynamic GHG inventories is namely to not assume when the peak occurs, but to do the assessment and to get peak temperature change and its timing as a result.

L374L: I don't understand why authors claim that delta T peak is a "non-H dependent" metric. It depends entirely on the lifetime of the gases emitted. This then determines the time horizon used for which delta T peak is defined, does it not?

There's some confusion in the reviewer's comment between lifetime (gas specific) and time horizon (user specific). Figure 4 shows that delta T peak is obtained at 11 year for a product and at 61 year for another product event same GHGs are present in the inventories.

L380-382: I do not understand why and how the authors think delta T long-term 'displays' the IPCC 2018 statement. More fundamentally, I don't understand why, given that the focus of this IPCC report was limiting warming to 1.5°C, which would occur in the 2050s, a long-term metric that only considers warming many centuries from now would help us achieve such a warming goal.

We agree that this statement is too simplistic. The original idea is that endpoint metrics stabilize much quicker than integrated metrics, i.e. better show that warming halt under specific conditions. We propose to better introduce the IPCC 1.5 SR statement:

" $\Delta T_{long-term}$  is representative of a stock climate change contribution of a product system, clearly showing that today's emissions will induce a rather stable long-term warming. This recalls that mitigation still leads to global warming, and only "reaching and sustaining net zero global anthropogenic  $CO_2$  emissions and declining net non- $CO_2$  radiative forcing would halt anthropogenic global warming on multi-decadal timescales", but not reduce it."

L400-402: I don't understand this sentence – its grammar seems incorrect

We agree and propose to remove the sentence

L420: I see no evidence in this manuscript why temperature end-point metrics are either complete or 'pragmatic policy choices' – especially given that the authors have not actually discussed any policy objectives, other than generic references to climate neutrality or carbon neutrality but without clarifying at what level – global, national, sectoral, product; and noting that in most national climate policy documents, such goals are clearly defined based on using GWP100.

We do understand the need for clarity. We propose to make clearer that we only speak of climate neutrality at a product system level. For instance, we propose to add:

in discussion: "Further research are needed to see the relevance of such metrics to evaluate yearly emissions at a sectoral, national or global level. "

in conclusion: "Proposed metrics particularly aim to reflect whether or not climate neutrality of product systems is achieved, either at short-, mid- or long-term perspectives depending on the policy objectives for which these metrics are applied. IPCC could support this by adopting AGTPpeak and AGTPlong-term, especially if their relevance to evaluate climate policies at national and global scales are confirmed by future research works."

L421: as per my general comments, predicting temperature is not a policy objective. The fact that delta T metric is about temperature does not mean it 'explicitly matches' the goal to limit global warming to (well!) below 2°C. Additional constraints have to be brought in to make such temperature prediction policy-relevant, such as a policy-relevant time horizon.

As proposed just above, proposed metrics enables to see at the same time whether a product system contribute to or mitigate global warming at short- or mid-term, how big is its peak temperature change, or whether or not it fits with long-term climate neutrality. It is the end user task to set a specific policy-relevant time horizon.

Nevertheless, we do agree to remove 'explicitly matches' and propose: "that more strongly resonate with the global target of halting and maintaining global warming below 2°C above the preindustrial level."

---

## Author Response (AR2)

**Referee #1**

Dear referee,

Thanks for noticing the improvement of the paper, while keeping some discussion points active. We understand that these points weren't satisfactorily addressed and propose further changes.

\_\_\_\_\_

Original comment \*45: The paper remains very unclear what "dynamic" means. The word is used 5 times in the paragraphs at the end of the Introduction, without any explanation. I provided a reference in my previous review ("for example Figure 8.30 of IPCC WG1 AR5 report and references therein") to what I thought "dynamic" means, but the authors did not adopt this, so I presume I am wrong. When reading the Glossary, I now maybe think that by "dynamic" they mean using a multi-pulse framework, but if this is so, why isn't this clearly stated? The new addition to the glossary "dynamic GWP and GTP are not included in this terminology" is an opaque statement. If this paper is intended to serve a pedagogical purpose to the LCA community, then they need to be much clearer.

[Authors]: Belonging to the LCA community, cite and shortly explain the major contribution of Levasseur et al. (2010) about dynamic LCA appeared enough. Though we agree to integrate your suggestion by adding:

"Hence, dCCA is based on a multi-pulse and multi-GHGs framework, as well as on absolute and dynamic climate impact assessment methods."

Also, in the glossary, we propose to erase "dynamic GWP and GTP" and to add:

"Here, relative metrics are not included in this terminology. This paper addresses in particular two dynamic climate metrics: cumulative radiative forcing ( $\Delta F$ ) and global mean temperature change ( $\Delta T$ )."

Original comment \*274: I originally pointed out "given that the time of peak warming varies little between gases, I can't see that AGTP\_longterm serves a usefully different purpose to AGTP(500)". The authors' response (that it is me that is confused) completely baffles me. When it is introduced in the paper it is absolutely clear that it is in the single t=0 pulse context, but their response clearly refers to the multi-purpose framework, which is an irrelevance at this stage in the paper, and likely an irrelevance when it is applied (as no evidence to the contrary is presented). I maintain that AGTP\_longterm and AGTP(500) will differ by such a small amount that introducing AGTP\_longterm introduces unnecessary complexity without any greater utility. I also remain of the contention that AGTP\_peak, if it has utility, could be adequately served by adopting a 10- or 15-year timescale, given the similarity of the AGTP\_peak amongst gases.

[Authors]: We originally wanted to have similarity in terms of recommendations between AGTP and  $\Delta T$  and got a bit stuck with our idea. We do now clearly understand your point, agree with it and propose to replace AGTPlong-term by AGTP500 in order to have a consistent reasoning at this stage of the paper, i.e. the single-pulse framework:

"In a single-pulse framework,  $AGTP_{500}$  appears to be a representative mean value of this observed temperature change flattening on a long-term perspective."

ΔTlong-term is now described in the multi-pulse framework only, latter in part 4 Results:

"In a multi-pulse framework, peak temperature change of product systems might occur decades (see Fig.4), or even centuries after  $t_0$ , which significantly shifts the time when temperature change becomes rather stable in a long-term perspective. Hence, instead of  $\Delta T_{500}$ , we propose the metric  $\Delta T_{long-term}$  being 500 years after  $\Delta T_{peak}$  in order to stay representative of the long-term temperature change flattening."

As for peak warming, we stay with our nomenclature proposal and propose to add peak timing in the indicator as suggested by the other referee.

Original comment \*324: In the previous version I was concerned about the authors claim that IPCC's AR6 discussion on metrics lacks clarity, and that it was unnecessarily negative about IPCC's presentation of the metric formulation. My contention was (and remains) that it is "only" the two-term climate response that is relatively hidden in AR6. The authors now state that in Smith et al. (2021) the CH4 and N2O "indirect effects are not recalled". While technically true, Smith et al. (2021) is the Supplementary Material to Forster et al. (2021) and in Forster et al. (2021) the indirect formulations are very clearly stated in Section 7.6.1.3. This apparently general criticism of IPCC is completely inappropriate and must be corrected.

[Authors]: The intention wasn't to criticize AR6 WG1 Chapter 7 as a whole. It was to notice that it requires a lot of energy and comprehension for a non-climate scientist to find all metrics equations, indirect effects particularities, parameter values and associated uncertainties. This observation actually motivated this paper.

Though, we do see now that it can be interpreted as a general criticism of IPCC and then propose to completely change the narrative.

"Supplementary Material of AR6 WG1 Chapter 7 has some limitations: [...]" is replaced by :

"Based on Forster et al., (2021) and Smith et al., (2021), this paper summarises main climate metric equations with no hidden parameter values: fast and slow response relaxation time values, as well as ECS value are explicitly given, as are their associated uncertainties; CH₄ and N₂O indirect effects are explicitly transcribed into metrics equations; CCf analytical solution is calculated and proposed in an open-access code page."

**Referee #2**

Dear referee,

By considering your comments, the paper gained indeed in clarity, consistency. Thank you again for that.

The association between the magnitude of  $\Delta$ Tpeak and its time occurrence is a discussion we also add. Your proposal to "include a requirement (or at least a suggestion) that the  $\Delta$ Tpeak reports the time at which the peak warming occurs, relative to t=0" is well-motivated. We do agree to support this by modifying the manuscript.

We propose to change Tables 2, 3 and S5 accordingly as well as to add in the main text:

"In this case [multi-pulse framework], peak timing is a required extra information. Hence, this metric that indicated both peak magnitude and timing occurrence appears even more pertinent."

As for the relevant suggestion to better reflect product system mitigation objectives, it will be considered in coming research work that will apply the methodology described here to case studies.

---

## Author Response (AR3)

Dear Editor,

Thanks for your encouraging response. In accordance with changes done in part 5.1, we agree to adopt your suggestion for the Introduction by replacing:

"First, IPCC does not provide the needed information in its latest report to easily understand and use dynamic climate metrics."

by

"First, the information on dynamic climate metrics provided in the latest IPCC report may be complex to use for those not trained in climate science".

Best regards, Vladimir Zieger and co-authors